# Impact of Thyroid Cancer Treatment on Renal Function: A Relevant Issue to Be Addressed

**DOI:** 10.3390/jpm13050813

**Published:** 2023-05-11

**Authors:** Rossella Di Paola, Ananya De, Anna Capasso, Sofia Giuliana, Roberta Ranieri, Carolina Ruosi, Antonella Sciarra, Caterina Vitagliano, Alessandra F. Perna, Giovambattista Capasso, Mariadelina Simeoni

**Affiliations:** 1Department of Mental and Physical Health and Preventive Medicine, University of Campania “Luigi Vanvitelli”, 80138 Naples, Italy; 2Department of Oncology, Livestrong Cancer Institutes, Dell Medical School, The University of Texas, Austin, TX 75063, USA; 3Nephrology Unit, Department of Specialist General Surgery, University Hospital “Luigi Vanvitelli”, 80131 Naples, Italy; 4Department of Oncologic Surgery, Translational Medical Sciences at University of Campania “Luigi Vanvitelli”, 80131 Naples, Italy; 5Nephrology and Dialysis Unit, Department of Translational Medical Sciences at University of Campania “Luigi Vanvitelli”, 80131 Naples, Italy; 6Biogem S.c.a.r.l. Research Institute, 83031 Ariano Irpino, Italy

**Keywords:** thyroid cancer, thyroid hormones, renal function biomarkers, cystatin C, NGAL, eGFR, cisplatin, radioiodine, immunotherapy, BRAF, BRAF inhibitors

## Abstract

Thyroid cancers require complex and heterogeneous therapies with different impacts on renal function. In our systematic literature review, we analyzed several aspects: renal function assessment, the impact of radiotherapy and thyroid surgery on kidney functioning, and mechanisms of nephrotoxicity of different chemotherapy, targeted and immunologic drugs. Our study revealed that the renal impact of thyroid cancer therapy can be a limiting factor in all radiotherapy, surgery, and pharmacological approaches. It is advisable to conduct a careful nephrological follow-up imposing the application of body surface based estimated Glomerular Filtration Rate (eGFR) formulas for the purpose of an early diagnosis and treatment of renal failure, guaranteeing the therapy continuation to thyroid cancer patients.

## 1. Introduction

Thyroid cancers, including papillary, follicular, and anaplastic variants, represent a growing health problem, accounting for more than 90% of all thyroid neoplasms within the differentiated thyroid cancers (DTCs). Although the prognosis of the latter is mostly favourable, the rate of recurrence is quite high [1]. Total/subtotal thyroidectomy with or without lymphadenectomy and radioactive iodine (I^131^) administration represents the standard of care of DTCs limited to the thyroid and is often complicated by dysthyroidism development. The therapeutic approach to recurrent and metastatic thyroid cancers is instead based on several aspects, such as the initial treatment, the site, and the disease extension [2]. In many cases of advanced thyroid cancers, chemotherapy is the only viable option, with a variable impact on renal function [1].

A physiological and pathological cross-link between thyroid and kidney has been widely described in the literature and is based on multiple hemodynamic and non-hemodynamic mechanisms [3,4]. Thyroid dysfunction directly affects kidney function by inducing a series of organ dysfunctions. In fact, dysthyroidism is one of the risk factors for the onset of chronic kidney disease. Recent research shows that overt thyroid disease is associated with significant changes in the main parameters for assessing kidney function (creatinine, glomerular filtration rate) [4]. The occurrence of acute or chronic kidney insult in thyroid cancer patients represent a fearsome complication, associated with quality-of-life worsening and increased hospitalization and mortality rates [5].

Our review is aimed at describing how different therapeutic approaches to thyroid cancers can affect renal function. For each treatment, the renal effects will be described, and any possible clinical advice will be provided. We will specifically analyse total and subtotal thyroidectomy, radioiodine therapy, chemotherapy, biological agents, and immunotherapy. The reliability of different renal function biomarkers in patients with thyroid cancer will be discussed. 

## 2. Materials and Methods

### 2.1. Conduct of Review

All relevant scientific articles were included in the analysis using a structured and methodical search approach, conducted in accordance with the Preferred Reporting Items for Systematic Reviews and Meta-Analyses (PRISMA) criteria and the International Prospective Register of Systematic Reviews (PROSPERO) guidelines.

### 2.2. Search Strategy and Study Selection

We searched the available literature focusing on the renal impact of different therapeutic approaches to thyroid cancers. Studies examining this topic were detected by a computerized research of all English-language articles in major electronic databases (Medline, PubMed, NIH, Cochraine, UptoDate, etc.). We carried out a systematic search of quality full-text papers by combining the following Medical Subject Heading (MeSH) terms: “Thyroid cancer and kidney”, “Thyroid cancer therapy guidelines”, “Renal function biomarkers in thyroid cancer and dysfunction”, “Thyroidectomy and renal function”, “Radioiodine and kidney”, “Radioiodine and end-stage renal disease”, “Radioiodine and dialysis”, “I^131^ and renal function”, “Cytotoxic chemotherapy in thyroid cancer”, “Cytotoxic chemotherapy in thyroid cancer and renal function”, “Cisplatin in thyroid cancer and renal function” “Doxorubicin in thyroid cancer and renal function”, “immunotherapy and thyroid cancer”, “targeted therapies in thyroid cancer”. Four hundred and seventy-two references were initially retrieved. One hundred-three references were excluded because they were not pertinent to our topic. Sixty-eight papers were discharged after full text analysis because there was no information on renal function in the studied populations. Two hundred forty-three papers were also discharged as they were not written in English or because they were abstracts, unpublished author manuscripts, and repository manuscripts. Finally, 22 papers were included in our analysis (Figure 1).

## 3. Renal Function Biomarkers

Renal function decline in thyroid cancer patients is a common complication that can be induced by both thyroid impairment and different oncologic therapeutic approaches. In all cases, it appears mandatory that renal function is assessed in a reliable way to drive the therapy in a safe and effective manner. Although novel biomarkers of renal function are available, GFR (Glomerular Filtration Rate) surrogate markers, such as Serum Creatinine (SCr), urine output, and creatinine-based estimating equations, are mostly used in clinical practice. However, creatinine could be influenced by the nutritional status and muscular mass loss in cancer patients and conventional eGFR formulas could be inaccurate. The correction by the body surface will consequently enhance their diagnostic power in thyroid cancer patients [6].

Cystatin C (Cys-C) is considered a highly reliable renal function parameter in the general population. However, being influenced by cellular turnover, its diagnostic power could be weakened by an altered thyroid status and by neoplastic diseases. In a recent report by Greco M. et al., cystatin C levels results were altered in hypothyroid patients on hormonal substitutive therapy with suboptimal Thyroid-Stimulating Hormone (TSH) levels despite CKD-EPI eGFR results being normal [3]. Similarly, cystatin C was found to be aberrantly high in a study conducted by Jones M. et al. on cancer patients [7], and the authors concluded that cystatin C could bias renal function evaluation in this patient’s typology for which a parameter to be strictly monitored in relation to the treatment is strongly needed.

Neutrophil Gelatinase-Associated Lipocalin (NGAL) is a 25 kDa glycoprotein whose expression in renal tubular cells, immune cells andcancer cells increases in specific clinical settings, such as acute kidney failure (AKI), inflammation, and infections. Because of that, NGAL is currently considered one of the most accurate early biomarkers of renal impairment [8,9,10]. However, this assumption is not as valid in thyroid cancer patients. Iannetti A. et al. reported that NGAL, being regulated by the Nuclear Factor kappa-light-chain-enhancer of activated B cells (NF-kB), is a survival factor for neoplastic thyroid cells and its levels could thus be influenced by thyroid cancer lesions, disrupting its diagnostic power [11].

In conclusion, thyroid cancer is a condition associated with an elevated risk of renal complications and based on our literature evaluation, it can be concluded that eGFR corrected by the body surface is the most reliable renal function parameter to be explored in thyroid cancer patients (Figure 2).

## 4. Total/Subtotal Thyroidectomy

The standard of care of DTCs is represented by total/subtotal thyroidectomy. It is often followed by hypothyroidism onset that requires substitutive hormonal therapy. In this condition, renal function decline might be a frequent complication. AKI is accompanied by increased health costs and adverse outcomes, including the progression of chronic kidney disease (CKD) and death [4,9,12]. In a retrospective study, it has been reported that AKI occurs in 4.8% of patients after non-cardiac surgical interventions [13]. 

In a study conducted on 486 patients, the incidence of post-thyroidectomy AKI was 4.9%. Multivariate logistic regression analysis showed that the risk factors associated with the occurrence of post-operative AKI were male gender, preoperative beta-blocker use, low preoperative serum albumin levels and colloid administration [14]. In patients with one or more of these risk factors, the renal function should therefore be closely monitored before proceeding with thyroidectomy. More attention should be paid to candidates for thyroidectomy with CKD whose AKI risk is even higher.

Kreismann et al. found that serum creatinine levels were elevated within 2 weeks after the onset of post-operative hypothyroidism. Although serum creatinine levels normalize after thyroid hormone replacement, recovery appears to be slower in patients with severe hypothyroidism [15]. Furthermore, renal hemodynamic changes in severe hypothyroidism have been observed to affect mainly glomerular rather than tubular function [16], and the activation of pro-inflammatory and pro-fibrotic pathways could lead to the onset of chronic kidney damage. 

The cross-link between thyroid and kidney is complex. Thyroid hormones influence renal function via direct and indirect effects on renal hemodynamic and intrarenal Renin-Angiotensin-Aldosterone System (RAAS) up-regulation [17]. The latter mechanism is involved in both AKI and chronic kidney damage. Hypothyroidism can induce renal function worsening through non-hemodynamic RAAS activities linked to pro-inflammatory and pro-sclerotic pathways. TSH also modulates Adenosine (ADO). In specific, a decreased parallel of Adenosine TriPhosphate (ATP) and ADO production was reported in hypothyroid experimental animals, with limited binding ability to the vasoconstrictive ADO-1 receptor. This leads to increased intracellular calcium in mesangial cells, renin release, afferent arteriole constriction, impaired urinary acidification, altered urinary osmolality, and decreased eGFR [4] (Figure 3).

Based on this evidence, an accurate renal function evaluation before and after thyroidectomy appears particularly important. The application of normalized to body surface eGFR equation is advisable [3]. 

## 5. I^131^ Radiotherapy

Therapy with I^131^ is recommended for most patients with DTC after total thyroidectomy [18]. It is approached to remove any residual thyroid cells that could not be removed by surgery.. Radioiodine ablation is also recommended in metastatic patients or in those with locally advanced or extensive (lesion diameter >4 cm) thyroid cancer. Patients with cancer elevated risk features should be ablated by I^131^ even if the primitive lesion is smaller than 4 cm and confined to the thyroid. Conversely, radioiodine ablation is not recommended in thyroid cancer patients with unifocal lesions <1 cm in the absence of other high cancer-risk criteria [19].

I^131^ is orally administered and a dose of 30 to 100 mCi is recommended in patients with low-risk lesions, while doses ranging from 100 to 200 mCi should be used in patients with a higher cancer risk [19,20]. It has been reported that in patients with CKD, iodine elimination is slowed with a possibly toxic increase in radioiodine levels in the body. In healthy subjects, 56% of radioiodine dose is excreted within 24 h, while in patients with CKD after one day only 11% of the dose is eliminated [21]. This leads to a three-and-a-half-fold accumulation of iodine in patients with CKD than in patients without the disease [22]. Dialysis only partially alleviates this problem, as I^131^ clearance is still much inferior in patients undergoing hemodialysis than in those without disease. For these reasons it would be necessary to reduce the dose of radioiodine administered to patients with CKD.

In this regard, some authors suggest doses of I^131^ between 5 and 30 mCi, slightly lower than in patients without CKD with low-risk lesions [23]. In addition to dose reduction, another strategy to reduce radioiodine nephrotoxicity would be to review the time interval between treatment and hemodialysis session. Some authors recommend a hemodialysis session within 48–72 h, others instead within 24 h after treatment with I-131 [24]. We retain that even residual renal function should be considered for choosing the right timing and possibly the hemodialysis session should be timely delivered and prolonged specially in anuric hemodialysis patients.

Radioiodine therapy is generally started within 1–3 months after thyroidectomy, when a TSH level over 30 mIU/L [25] is achieved by either thyroid replacement therapy discontinuation or recombinant human TSH (rhTSH) infusion. TSH increase will, in fact, enhance the thyroid ablative power of I^131^ through the increased avidity of thyroid for iodine caption. Based on the robust knowledge on a significant cross-link between thyroid and kidney, particular attention should be paid to the risk of GFR decline due to the induced dysthyroidism [26]. In this regard, TSH is responsible for modulating the vasoactive molecule adenosine (ADO). ADO, a metabolite of adenosine triphosphate (ATP), binds the ADO A1 and ADO A2 receptors with high affinity, inducing a series of cascading effects responsible for vasoconstriction and vasodilation. A reduction in ATP production has been observed in hypothyroid in vivo models, which directly induces a reduction in ADO expression, and consequently a reduced activation of the vasoconstrictor receptor ADO-1. The binding of ADO and ADO-1 induces several glomerular effects, including increased renin release, altered urinary acidification, altered urinary osmolarity, constriction of afferent arterioles and reduced GFR [4].

Yoon Yang Cho et al. have compared the impact on renal function of the two radioiodine sensitization modalities (discontinuation of hormone therapy and rhTSH infusion) and demonstrated that a better renal rescue was achieved with rhTSH infusion, for which the renal disruption is shorter than in hormonal therapy discontinuation. Moreover, rhTSH allows background thyroid hormonal therapy maintenance with better post-radioiodine hypothyroidism compensation. Thus, in thyroid cancer patients with impaired renal function at baseline, the use of pre-radioiodine rhTSH could be recommended [27].

Radioiodine therapy efficacy is also mediated by the Na^+^/I^−^ Symporter (NIS) activity, an integral plasma membrane glycoprotein located at the basolateral plasma membrane of thyrocytes, where it mediates active iodine accumulation into thyroid follicular cells [28]. NIS was thought to be expressed only in the thyroid; however, it has also been found in the kidney. Spitzweg et al. reported a significant NIS expression in the distal tubule, while it was only slightly represented in the proximal tubule and absent in the glomerulus [29]. Because of that, we would raise attention to the possible renal toxicity of radioiodine therapy. Aktoz et al. tested radioiodine therapy on albino rats and reported tubular damage at the five-day follow-up after a single radioisotope dose [30]. Of note, in CKD patients, even a body iodine accumulation can occur due to their impaired ability to excrete the ion [31] with possible reflexes on both thyroid and renal function. A 50% I^131^ dose reduction has been suggested in patients on haemodialysis. It is recommended that each I^131^ infusion should be preceded and followed at 48 h by a haemodialysis session [32,33]. In patients on peritoneal dialysis, the clearance of I^131^ was found to be less than 20% compared to normal renal function patients [34], suggesting a proportional dose reduction even in this patient typology. Indeed, more investigations would be needed to address this issue. 

In conclusion, therapy with radioiodine is indicated only in intermediate and high-risk thyroid cancer. Thyroid sensitization to radioiodine increases radiotherapy effectiveness, however it can lead to an eGFR decline in CKD patients. Thus, a radioiodine dose reduction and the use of rhTSH is indicated in these patients. Moreover, in patients on hemodialysis a revision of both the dialytic scheme and radioiodine dose is recommended.

## 6. Chemotherapy

### Conventional Drugs

The use of conventional chemotherapy in thyroid cancer patients is extremely limited since too little data exist to recommend specific cytotoxic regimens. According to the 2015 American Thyroid Association Management Guidelines for Adult Patients, cytotoxic chemotherapy is indicated in high-risk metastatic radioiodine refractory thyroid cancer patients not eligible for other therapeutic approaches [19].

Doxorubicin and Cisplatin are the most used cytotoxic drugs, both with potentially nephrotoxic side effects. Moreover, Cisplatin and Doxorubicin are commonly burdened by gastrointestinal side effects, such as vomiting and diarrhea, that could also trigger dehydration related AKI, especially in patients with previous renal damage [19].

Nephrological follow-up is therefore recommended for all patients with thyroid cancer undergoing cytotoxic chemotherapy. Serum and urinary electrolytes, 24 h proteinuria, acid-base balance should also be monitored for tubule-toxicity risk monitoring [35].

Doxorubicin belongs to the anthracyclines family and its administration is aimed at sensitizing locally advanced thyroid tumours to radioiodine response. However, the risk-benefit balance is still unknown [19]. In children with rapidly progressive, symptomatic, and/or imminently life-threatening thyroid cancer, Doxorubicin is used either as a single agent or in combination with Cisplatin or Interferon-α [36]. However, the drug effectiveness even in paediatric advanced thyroid cancers appears extremely poor. Although quality data would be needed, no clinical trials have been conducted in children with I^131^refractory thyroid cancer so far.

In addition to this limited availability of data on Doxorubicin efficacy profile, its nephrotoxicity might be another relevant issue limiting the use of the drug in thyroid cancer patients with CKD. Aiping Li et al. studied the time-dependent changes of urinary metabolomics in rats undergoing treatment with Doxorubicin. Butanoate, alanine, aspartate, glutamate, arginine, and proline metabolic cascades were recognized as important pathways associated with Doxorubicin-induced nephropathy. Thus, the authors concluded that the urinary metabolomics study is a promising method not only to elucidate toxicological mechanisms underlying Doxorubicin-induced nephrotoxicity but also to detect early changes before parenchymal organ injury. However, confirmation in humans would be needed [37].

A wider knowledge exists for Pegylated Liposomal Doxorubicin (L-Doxo): it has scarce cardiac uptake and a prolonged half-life with improved both anticancer effectiveness and safety profile compared with non-lysosomal formula. Even myelosuppression, vomiting, and alopecia are more infrequent during L-Doxo administration. However, cases of AKI, nephrotic syndrome and renal thrombotic microangiopathy have been described in patients with long-term L-Doxo treatments. It should be emphasized that these renal complications could occur even several months after drug discontinuation, and GFR restoration is not surely observed [38,39,40].

A L-Doxo related renal cell damage has been demonstrated with increased glomerular capillary permeability and tubular atrophy [41]. L-Doxo triggers apoptotic cascades through mitochondrial and death receptor pathways [42]. In addition, recent studies confirmed that L-Doxo can stimulate various pro-inflammatory mediators such as Cyclooxygenase-2 (COX-2) and NF-κB [43] with a negative impact on both renal hemodynamics and tissue remodelling.

Cisplatin is an alkylating agent whose use in thyroid cancer is limited to the advanced disease in presence of a refractoriness to radioiodine treatment. It is usually associated with Doxorubicin, although nephrotoxicity is the most common cause of dose reduction and/or treatment suspension. In clinical practice, the overall prevalence of Cisplatin-induced nephrotoxicity occurs in one third of treated patients and approximately ten days after treatment.

Cisplatin is eliminated by the kidney and can accumulate in renal proximal tubules. As for that, its use is generally limited to patients with eGFR > 60 mL/min. However, with Cancer Care Ontario endorsement, Kintzel et al. suggested, respectively, a 25% and a 50% dose reduction in cancer patients having a KDIGO stage IIIa CKD (eGFR 60–46 mL/min), and a KDIGO stage IIIb CKD (eGFR 45–30 mL/min) [44]. Conversely, Bennet et al. proposed the use of Cisplatin even in cancer patients with lower eGFR and suggested a 25% dose reduction in patients having a creatinine clearance ranging from 50 to 10 mL/min and a 50% dose reduction only if eGFR is below 10 mL/min [45]. Considering the additional renal insult due to dysthyroidism in DTC patients, we would better agree with Kintzel’s recommendation regarding Cisplatin use in CKD patients. A better consensus can be found on Cisplatin dose reduction in patients undergoing dialysis, although even this issue deserves more studies for confirmation. So far, a post-dialysis administration and a 50–75% dose reduction is recommended in haemodialysis patients. Peritoneal patients instead require a 50% dose reduction without any recommendation regarding the administration modality [45].

Cisplatin uptake by epithelial cells in the proximal renal tubule is driven by two basolateral transporters: the Copper TRansporter 1 (CTR1) and type 2 Organic Cation Transporter (OCT2). Both carriers have a high affinity for Cisplatin, conversely to the apical transporter called type 1 Multidrug and Toxin Extrusion (MATE1), which has a low affinity for the drug. This explains the tendency of Cisplatin to accumulate in tubular cells, where activation of all cytotoxic Reactive Oxygen Species (ROS), protein-53 (p53), and Microtubule-Associated Protein kinase/Extracellular Regulated Kinase (MAP-ERK) kinase cascades leads to mitochondrial damage and apoptosis [46,47].

These mechanisms explain the high incidence of AKI and Fanconi-like syndrome in patients treated with Cisplatin. Daniel J. Crona et al. reported the importance of ensuring adequate hydration in all patients undergoing Cisplatin [48]. Even magnesium supplementation (8–16 milliequivalents) may limit Cisplatin-induced nephrotoxicity [49,50], and Mannitol should be considered in patients undergoing high-dose Cisplatin and/or having preexisting hypertension [48].

In conclusion, although several studies report Cisplatin and/or L-Doxo nephrotoxicity in other than thyroid cancer, the current literature is insufficient to conclude that these drugs show an increased renal toxicity when used in thyroid cancer patients. More investigations on this important topic would be thus desirable. However, based on our review, it appears that the use of conventional chemotherapeutics in patients with advanced thyroid cancer could be more difficult since a dysthyroidism at baseline could increase these drugs’ nephrotoxicity. Different strategies, such as drug dose reduction, hydration cycles, magnesium supplementation and/or mannitol administration, should be then carefully evaluated in thyroid cancer patients.

## 7. Biological Agents

Knowledge of thyroid cancer biology has steadily increased in recent years with a rapid evolution of the diagnostic and therapeutic approach to the disease. Tyrosine Kinase Inhibitors (TKI), Mitogen-activated Extracellular signal-regulated Kinase (MEK) and B- Rapidly Accelerated Fibrosarcoma (BRAF) kinase inhibitors and immunotherapy are currently used for treating thyroid cancer patients.

Cancer biology studies have revealed a certain degree of intratumoral heterogeneity within thyroid cancer typologies, highlighting the coexistence of cellular subpopulations with distinct proliferative capacities and differentiation abilities. Cancer Stem-like Cells (CSCs) are hypothesized to contribute to the metastatic potential and therapy resistance of thyroid cancer. CSCs principally exist in tumor areas with specific micro-environmental conditions, the so-called stem cells niches. This evidence has suggested the use of different molecules that directly target the pathways necessary for CSC survival, alone or in combination with conventional anticancer drugs [51].

Targeted therapies represent an important therapeutic option for the treatment of all advanced cases of radioiodine refractory DTC, Medullary Thyroid Cancer (MTC), and cases of Poorly Developed Thyroid Cancer (PDTC) and Anaplastic Thyroid Cancer (ATC). Many novel biological agents used in clinical practice still lack specificity and selectivity and have the propensity to inhibit multiple targets. The biological consequences of multi-kinase activity are poorly defined, and numerous class-specific toxicities have been described. The kidney is an organ in which most of these targeted pathways are expressed. Preclinical data and human kidney biopsies have provided insight into the mechanisms involved in how targeted agents can cause kidney toxicity [52].

### 7.1. Tyrosine Kinase Inhibitors (TKI)

Tyrosine Kinase Inhibitors (TKIs) represent the most studied pharmacological class for thyroid cancer treatment, as they inhibit different pathways that participate in either cell proliferation or stemness acquisition. The most studied TKI agents are the inhibitors of the Endothelial Growth Factor Receptor (EGFR) pathway. In fact, quiescence, glycolytic metabolism and immunosuppressive activity are modulated by EGFR and its downstream target Signal Transducer and Activator of Transcription 3 (STAT3) in CSCs [52].

In recent years, several TKIs have been evaluated for the treatment of advanced, progressive and radioiodine resistant thyroid cancers, and some of them have recently been approved for use in clinical practice: Sorafenib and Lenvatinib for DTC and PDTC; Vandetanib and Cabozantinib for MTC.

TKIs show several renal effects. It has been reported that they can induce hypocalcemia [53] and hypophosphatemia by acting directly on proximal tubular cell receptors [54]. Proteinuria is the main renal side effect of TKIs, and it occurs mainly in hypertensive patients with diabetes [55].

Sorafenib was the first TKI agent to be approved by the Food and Drug Administration for the treatment of progressive metastatic thyroid cancer refractory to radiotherapy [56]. Sorafenib target Types 1–3 Vascular Endothelial Growth Factor Receptors (VEGFR 1–3), the Platelet-Derived Growth Factor Receptor (PDGFR), Rearranged during Transfection (RET), and the proto-oncogene c-KIT. It has been reported as a safe and effective drug for the treatment of CKD patients. No dose adjustments are recommended in these patients, although it would be advisable to start with Sorafenib at a lower initial dose followed by drug titration up to the maximum tolerated dose [57].

In a study conducted on a cohort of patients treated with Sorafenib, a group having baseline renal dysfunction (eGFR > 60 mL/min/1.73 mq) compared to the controls (eGFR < 60 mL/min/1.73 mq) showed a stable renal function during the treatment. AKI, hypertension and proteinuria had similar incidences in the two groups and occurred at low frequency. However, several adverse events occurred in both groups during the 12-month observation period leading to discontinuation or dose reduction [58].

Lenvatinib is another TKI approved in 2015 for the treatment of progressive DTCs refractory to radioiodine [59]. Lenvatinib targets VEGFR2, VEGFR3, EGFR, PDGFR, RET, and c-KIT. It is more effective but also more toxic than Sorafenib, with no easy drug management. Proteinuria and renal failure, along with QT interval prolongation and fatal tachyarrhythmias have been reported among the most frequent Lenvatinib induced adverse events, often leading to dose interruptions or changes [60].

Zhang et al. observed that the most common renal adverse event of Lenvatinib is proteinuria [61]. Even in a phase-three study, proteinuria was reported as the main adverse event leading to dose reduction and discontinuation of Lenvatinib treatment. Proteinuria occurs approximately 6 weeks after the start of therapy [62], and it takes approximately 9 weeks to resolve after Lenvatinib discontinuation or dose reduction [63]. Steroids are a useful therapeutic approach aimed at the treatment maintenance in proteinuric patients with elevated risk of thyroid cancer progression [60].

In CKD patients, a lower starting dose of Lenvatinib (i.e., 14 mg/die instead of 24 mg/die) is recommended with possible drug up-titration during treatment.

A study comparing the renal impact of Sorafenib vs. Lenvatinib was conducted on 73 patients with radioiodine refractory thyroid cancer. Proteinuria and eGFR decline were more frequent with Lenvatinib than Sorafenib. However, Lenvatinib dose reduction was mostly followed by a resolution of renal toxicity, allowing anticancer treatment continuation [64].

Interestingly, there is also a published case report of serial switches from Lenvatinib to Sorafenib. A 56-year-old patient with radioiodine refractory thyroid cancer developed nephrotic syndrome at 1 month-follow-up after Lenvatinib 20 mg/day initiation. Lenvatinib dose reduction to 10 mg/day was not able to resolve the nephrotic syndrome. Conversely, the Lenvatinib substitution with Sorafenib 400 mg/day induced a complete remission of the nephrotic syndrome in the absence of cancer progression at 5-month follow-up [65].

In conclusion, Lenvatinib shows higher effectiveness but also more nephrotoxicity in patients with DTC. Based on the available literature, it appears that a careful Lenvatinib dose titration or a switch to Sorafenib might solve the nephrotoxicity without losing the therapeutic goal.

### 7.2. MEK and BRAF Kinases Inhibition

Another recent therapeutic approach to radioiodine refractory thyroid cancer relies on the upregulation of NIS, via the inhibition of its negative regulators MAP and BRAF kinases. This strategy re-enables the incorporation of radioiodine in cancer cells and is based on pre-treatment with kinases inhibitors [66].

The landmark study by Ho et al. has proven the principle that the inhibition of MEK1 and MEK2 by Selumetinib induces radioiodine uptake in thyroid cancers refractory to radiotherapy [67].

The MAPK signaling cascade, including reticular activating system (RAS), RAF, MEK, and ERK, regulates cell proliferation, differentiation, apoptosis, and survival [68]. In 80% of patients with Papillary Thyroid Cancer (PTC), activating mutations are identified in genes that code for signaling molecules within the MAPK pathway [69].

The constitutive activation of the homologous BRAF by a somatic mutation causes oncogenic transformation of normal cells. Mutations in the isoform RAF-BRAF typically involve the replacement of valine with glutamic acid at the amino acid residue 600 (V600E) and are observed in 36–86% of patients with PTC and in 20–25% of patients with Anaplastic Thyroid Carcinoma. In these patients, the BRAF V600E mutation is associated with a worse prognosis, including higher clinical stage, tumor size, extra-thyroid extension, lymph-nodal metastases, tumor recurrence, and radiotherapy refractoriness [69].

In BRAF-mutant PTC cell lines, selective targeting of BRAF V600E inhibited proliferation, decreased ERK and MEK phosphorylation, induced G block and impaired the expression of genes involved in controlling the GS cell cycle transition. Similar promising results were observed in a BRAF-mutant PTC xenograft model, in which the selective targeting of BRAF V600E achieved tumor inhibition, as well as reduction in phospho—ERK and phosphor—MEK [69].

Dabrafenib, a potent ATP-competitive inhibitor of BRAF kinase, is selective for mutant BRAF in kinase panel screenings, cell lines, and xenografts. In a study by Abbas et al., a Dabrafenib-associated GFR decline was reported in 25% of patients and 10% of them had interstitial nephritis [70].

Adverse events associated with the treatment with BRAF inhibitors include renal events, such as allergic interstitial nephritis, isolated electrolyte disturbances (hypophosphatemia, hyponatremia, hypokalemia), and sub-nephrotic proteinuria and acute tubular toxicity that occurred in the early phase of drug treatment [71].

In a case report published by Launay-Vacher et al., it was observed in a cohort of eight patients that treatment with BRAF inhibitors caused a decrease in gromelular filtration. One of the patients underwent renal biopsy, which showed acute tubular necrosis. In total, 35% of the patients recovered renal function after discontinuation of the drug while 25% despite discontinuation of therapy did not recover their baseline renal function [72]. Another study showed that about 3 months after starting treatment with BRAF inhibitors 78% of the patients developed acute renal failure. Renal biopsies showed tubular toxicity and interstitial fibrosis [73].

Renal failure was diagnosed in <1% of patients treated with BRAF inhibitor. Flaherty et al. observed hyponatremia, hypophosphatemia, increased serum creatinine and hypokalemia in patients treated with a combination therapy of Debrafenib + MEK inhibitor [74].

BRAF inhibitor activity was also observed in a patient with end-stage renal failure. No worrisome side effects were observed in this case, but in any case, careful monitoring of electrolytes and cardiological parameters is recommended in dialysis patients treated with BRAF inhibitors [75].

Therefore, the monitoring of serum creatinine, electrolytes, and careful assessment of eGFR prior to initiation of therapy is recommended during treatment with BRAF and MEK inhibitors. Particular attention should be paid to the first months of therapy, which is when renal damage mainly develops.

## 8. Immunotherapy

Immune checkpoint inhibitors are innovative drugs whose use was recently approved for the treatment of advanced cancer. The impact of Ipilimumab (Cytotoxic T-Lymphocyte Antigen 4 (CTLA-4)-antagonist), Nivolumab and Pembrolizumab (Programmed cell Death protein 1(PD-1)-antagonists), and Atezolizumab (Programmed Death-Ligand 1(PD-L1)-antagonist) on overall survival in cancer patients is relevant. Thyroid tissue has a distinctive immunogenicity, and PD-L1 expression has been reported on thyroid cancer cells along with PD-1 on lymphocytes infiltrating thyroid tumors microenvironments. In DTC, an increased PD-1 expression is predictive of a more aggressive phenotype, while PD-L1 correlates with a higher recurrence risk. Anti-CTLA4 agents trigger the immune response of cancer since their PD-L1 (B7-H8) and PD-L2 (B7-DC) ligands are expressed in several tumor types [76,77,78]. Only a few studies have investigated immunotherapy in thyroid cancer. Only a few anecdotic cases of anaplastic thyroid cancer have been successfully treated with immune checkpoint inhibitors [79]. However, the association of Ipilimumab with Nivolumab has not been approved for DTC treatment so far.

Future studies would be necessary for assessing the efficacy and safety of this important drug class in DTC.

## 9. Discussion

Based on our literature review, we draw some conclusions and suggestions addressed at limiting the onset and progression of the kidney damage related to DTC therapy. Partial or total thyroidectomy is the gold standard therapy in localized DTC [80]. Its renal impact relies on the sudden drop in thyroid hormones levels in the blood [14]. It is thus advisable to monitor the normalized to body surface e-GFR before and after surgery, Cystatin C should not be considered as a reliable renal function biomarker in DTC patients at any disease stage [6]. A careful post-surgery hypothyroidism compensation is also strongly desirable, due to its widely known impact on renal function and kidney damage progression [81,82].

Adjuvant radiotherapy is indicated in intermediate to high-risk thyroid cancer. To achieve a better renal rescue without losing the therapy effectiveness, in DTC patients with impaired renal function at baseline, it is recommended the use of rhTSH instead of hormone therapy discontinuation [27].

A reduction of the empirical I^131^ dose should be considered in CKD patients, achieving a 50% in hemodialysis patients that should be treated 48 h before and after the administration of the radioisotope [32,33]. A similar dose reduction is also required in peritoneal dialysis patients [34].

As for patients eligible for Cisplatin and/or Doxorubicin, renal function should be carefully evaluated at baseline and during treatment. The normalized-to-body surface CKD- EPI equation remains the best diagnostic tool in this patient typology [6]. Before treatment initiation, it is recommended to ensure adequate hydration and to correct intravascular depletion to maintain renal perfusion [48]. Attention should be paid to avoid the concomitant use of nephrotoxic drugs whenever possible and to maintain drug levels within the recommended therapeutic range.

Iodinated contrast medium should only be used when strictly necessary, especially in CKD patients with concomitant diabetes, heart failure, metabolic acidosis, anemia, and dehydration [83].

In particular, to be safe and feasible in CKD patients, Cisplatin treatment should be supported with outpatient hydration regimens and have short-duration and low-dose. Moreover, Magnesium supplementation (8–16 mEq/kg) may limit Cisplatin-induced nephrotoxicity, and mannitol may be considered for high-dose Cisplatin and/or patients with preexisting hypertension. The monitoring of all serum and urinary electrolytes, 24 h proteinuria and acid-base balance is essential for early diagnosis and treatment of tubular damage in patients treated with Cisplatin [48].

Therapeutic approach to DTC is rapidly evolving and the use of TKI has markedly improved the prognosis [84]. The administration of oral steroids allows the management of autoimmune nephropathy without interrupting Lenvatinib [53]. TKI dose reduction also promotes the resolution of renal adverse events and successful DTC treatment continuation [64]. The replacement of Lenvatinib with Sorafenib may also be considered to reduce TKI nephrotoxicity [65].

## 10. Conclusions

In conclusion, DTC treatment, including surgery, chemotherapy, and targeted drugs, is associated with a significant survival improvement, but nephrotoxicity remains a limiting factor [85,86]. As for that, preliminary renal function assessment and personalized therapy and support is essential for allowing treatment completion (Table 1).

## Figures and Tables

**Figure 1 jpm-13-00813-f001:**
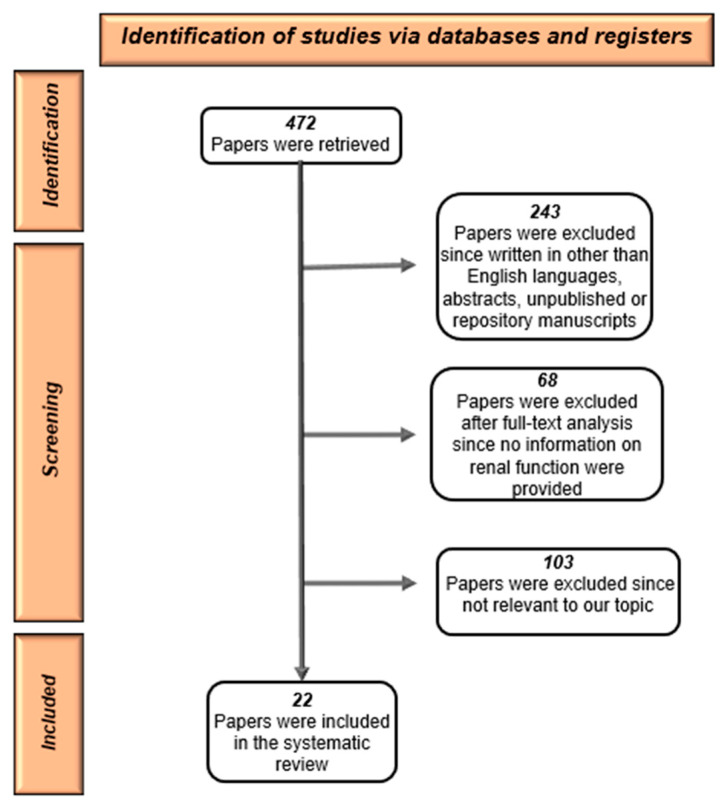
Flow chart of the literature selection process.

**Figure 2 jpm-13-00813-f002:**
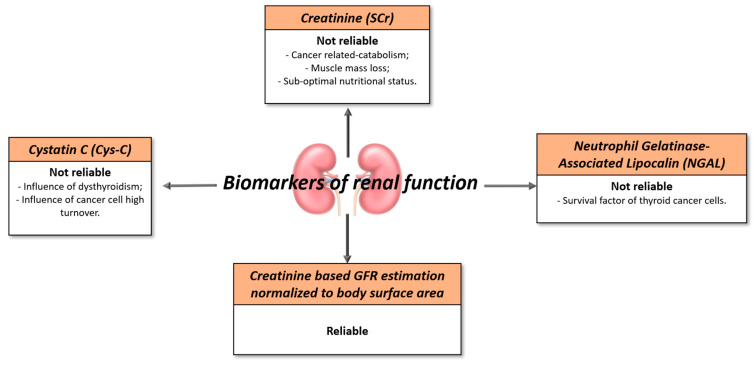
Limits and strengths of renal function biomarkers.

**Figure 3 jpm-13-00813-f003:**
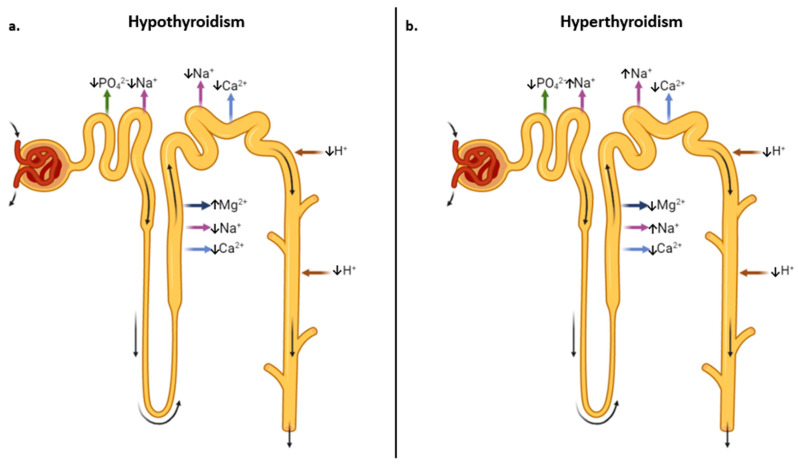
Physiological and-pathological cross-link between thyroid and kidney. Thyroid hormones directly influence various renal tubular functions. (**a**) Hypothyroidism is associated with hyponatremia (↓ renal Na^+^ reabsorption), hypercalciuria (↓ renal Ca^2+^ reabsorption), hyperphosphaturia (↓ renal PO_4_^2−^ reabsorption), hypomagnesuria (↑ renal Mg^2+^ reabsorption), and a tendency to distal tubular acidosis (↓ renal H^+^ excretion). (**b**) Hyperthyroidism is associated with sodium retention (↑ renal Na^+^ reabsorption), hypercalciuria (↓ renal Ca^2+^ reabsorption), hyperphosphaturia (↓ renal PO_4_^2−^ reabsorption), hypermagnesuria (↓ renal Mg^2+^ reabsorption) and distal tubular acidosis (↓ renal H^+^ excretion).

**Table 1 jpm-13-00813-t001:** Summary table of different thyroid cancer treatments associated with renal complications. ↑ increases; ↓ decreases. Abbreviations: AKI → Acute Kidney Injury; ESRD → End-Stage Renal Disease; ROS → Reactive Oxygen Species.

DCT Treatments	Reference	Renal Adverse Event	Study Design	Study Group	Intervention
**Surgery**	Joo EY et al. [14]	↑AKI incidence in post-surgical hypothyroidism	Retrospettive Observational	486 patients	Total thyroidectomy (81.5%), Hemithyroidectomy (16.0%), Complete thyroidectomy (2.5%)
	Kreismann SH et al. [15]	↑serum creatinine levels	Retrospective Observational	24 patients	Total thyroidectomy
**Radiotherapy**	Aktoz T et al. [30]	↑ tubular damage	Experimental	50 Wistar albino rat females	Treatment with Radioactive Iodine^131^
	Kaptein EM et al. [34]	↓ 20% I^131^ clearance in the patients with ESRD	Prospective Observational	-10 patients with thyroid carcinoma -8 patients with ESRD	Treatment with Radioactive Iodine^131^
**Chemotherapy**	Pabla N et al. [35]	↑ risk of tubulotoxicity ↓ glomerular filtration rate ↑serum creatinine	Systematic Review	Patients, cells and experimental animals	Treatment with Cisplatin
	Li A et al. [37]	↑nephropathy	Experimental	Male Sprague Dawley Rats	Treatment with Doxorubicin
	Yemm KE et al. [38] Shavit L et al. [39] Carron PL et al. [40]	↑AKI ↑nephrotic syndrome ↑renal thrombotic microangiopathy	Prospective Observational	Cohort of patients	Treated with L-Doxo
	Kintzel PE et al. [44] Karasawa T et al. [46]	↑activation of ROS, p53 and MAP-ERK kinase cascades in tubular cells ↑mitochondrial damage ↑apoptosis	Systematic Review	Patients, cells and experimental animals	Treatment with Cisplatin
**Target therapy-TKI**	Al-Jundi M et al. [59]	↑glomerular damage with secondary proteinuria ↑tubulo-interstitial damage without proteinuria	Systematic Review	Cohort of patients	Treatment with Lenvatinib
	Zhang W et al. [61]	↑proteinuria of any degree	Meta-analysis	9446 patients	Treatment with Lenvatinib
	Schlumberger M et al. [62]	↑proteinuria	Clinical trial	59 patients with thyroid carcinoma	Treatment with Lenvatinib
	Haddad RI et al. [63]	↑proteinuria	Clinical trial	392 patients	Treatment with Lenvatinib
	Iwasaki H et al. [64]	development nephrotic syndrome at 1-month follow-up	Case report	56-year-old man with DTC refractory	Treatment with Lenvatinib
**Target therapy-MEK and BRAF inhibitors**	Falchook GS et al. [69]	↓glomerular filtration rate ↑creatinine levels	Prospective Observational	14 patients with thyroid carcinoma	Treatment with Dabrafenib
	Wanchoo R et al. [71]	allergic interstitial nephritis hypophosphatemia, hyponatremia, hypokalemia, subnephrotic proteinuria acute tubular toxicity	Systematic Review	Cohort of patients	Treatment with BRAF and MEK inhibitors
	Launay-Vacher V et al. [72]	↓ glomerular filtration	Case reports	8 patients	Treatment with BRAF inhibitors
	Teuma C et al. [73]	↑creatinine levels	Retrospective Observational	74 patients	Treatment with BRAF inhibitors
	Flaherty KT et al. [74]	hyponatremia hypophosphatemia ↑serum creatinine and hypokalaemia	Clinical trial	162 patients	Treatment with BRAF and MEK inhibitors
**Immunotherapy**	Borówka M et al. [77]	Tubulointerstitial nephritis Nephrotic syndrome Tubulointerstitial nephritis	Systematic Review	Cohort of patients	Treatment with Ipilimumab

## Data Availability

No new data were created or analyzed in this study. Data sharing is not applicable to this article.

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
