# Peer review of "Impact of Thyroid Cancer Treatment on Renal Function: A Relevant Issue to Be Addressed"

_jpm, 2023, doi:10.3390/jpm13050813_

Round 1

Reviewer 1 Report

It is an interesting topic:

I could not see difference in the 2 sections of Fig 1, and if it has a reference, did it pass copyright?

Introduction needs to be expanded with sufficient information

BRAF inhibitors were not included in the keywords

Section 4: it is unclear, how surgery and renal function were connected. and what are recommendations the surgeon should do perioperative? and if there is impact of extent of surgery of thyroid hormone replacement onrenal outcomes? what timeline we expect thyroidectomy patients deteriorate or need special care?

Should prior renal patients have special care?

Section 5: how RAI is connected with renal function, does it complicates the situation, Should we adjust RAI dose based on prior renal function tests? Is there any prognostic indicator for renal outcomes based on patients' covariates and demographics?

Section 7: summarize thyroid treatment without connecting to renal function or outcomes.

Discussion: need references in half of paragraphs.

Author Response

We are deeply grateful to the Reviewers for their comments and suggestions that were clearly addressed at improving the quality of our manuscript. Here we provide the point-to-point answers to Reviewers’comments.

Reviewer 1

Reviewer: I could not see difference in the 2 sections of Fig 1, and if it has a reference, did it pass copyright?

Answer: Thanks for having pointed out this issue. Figure 1 was created ex-novo and is not a copyright. We agree that the presence of the citation in the figure can be misleading. We added the reference since the concept in the figure is described in that paper. In the new version, we have removed it the citation from the legend.(lines 173-182)

Reviewer: Introduction needs to be expanded with sufficient information

Answer: We agree with the reviewer. In the current version of the manuscript, we extended the introduction accordingly (lines 48-52; 57-61).

Reviewer: BRAF inhibitors were not included in the keywords

Answer: Undoubtedly, this was a it was a serious oversight. We added BRAF inhibitors to the keywords in the revised manuscript.

Reviewer: Section 4: it is unclear, how surgery and renal function were connected. and what are recommendations the surgeon should do perioperative? and if there is impact of extent of surgery of thyroid hormone replacement on renal outcomes? what timeline we expect thyroidectomy patients deteriorate or need special care? Should prior renal patients have special care?

Answer: We thank the reviewer for his suggestion. To clarify the association between surgery and renal function, we have added a new paragraph to section 4. In particular, we have described the possible thyroid status related risk factors for acute postoperative renal failure , the renal effects caused by surgical therapy and the aspects of the patient to be monitored before and after surgery(lines 143-156).

Reviewer: Section 5: how RAI is connected with renal function, does it complicates the situation, Should we adjust RAI dose based on prior renal function tests? Is there any prognostic indicator for renal outcomes based on patients' covariates and demographics?

Answer: We thank the Reviewer for raising this important issue. In the current version of the manuscript, we have included the differences in radioiodine treatment between patients with and without renal problems at baseline.(lines 196-211;220-228)

Reviewer: Section 7: summarize thyroid treatment without connecting to renal function or outcomes.

Answer: Thanks to the Reviewer for this very shareable comment. We have added for both tyrosine kinase inhibitors and MEK and BRAF inhibitors new information on the association of treatment with renal function. In addition, we have evaluated the possible measures necessary to ensure continuation of therapy taking into account the patient's health status.(lines:383-386;391-401;417-425;473-495)

Reviewer: Discussion: need references in half of paragraphs.

Answer: We are very grateful for having pointed out this major deficiency. Current manuscript has been extended several new citations belonging to the discussion.

Reviewer 2 Report

Impact of thyroid cancer treatment on renal function: a relevant issue to be addressed

the study is a comprehensive insight into the association of renal function and thyroid cancer.

I only have a couple of remarks:

Line 74 – “Cisplatin in thyroid cancer and renal function” “Cisplatin in thyroid cancer and renal function” – some repetitions

Line 116 – “so far from” – the manuscript needs some minor language corrections

Line 180 - albine rats – did you mean “albino rats”?

Line 318 – “Sorafenib (SOF)” please avoid using so many abbreviations, I think it is especially not useful to use abbreviations for single words, like in this line (it doesn’t lower the word count, but it can introduce confusion for the reader). Other example is LEN (Lenvatinib)

Author Response

We are deeply grateful to the Reviewers for their comments and suggestions that were clearly addressed at improving the quality of our manuscript. Here we provide the point-to-point answers to Reviewers’comments.

Reviewer 2

Reviewer: Line 74 – “Cisplatin in thyroid cancer and renal function” “Cisplatin in thyroid cancer and renal function” – some repetitions lines 75

Answer: We thank the reviewer for his suggestion. Yes there is a repetition, we have now eliminated it in the revised manuscript.

Reviewer: Line 116 – “so far from” – the manuscript needs some minor language corrections line 116

Answer: Thanks to the Reviewer for this very shareable comment. We have amended this point (line 127) and revised the entire article to improve English.

Reviewer: Line 180 - albine rats – did you mean “albino rats”? line 226

Answer: We agree with the reviewer, it was a mistake we mean albino rats. We have now replaced the incorrect term with the correct one.(line 243)

Reviewer: Line 318 – “Sorafenib (SOF)” please avoid using so many abbreviations, I think it is especially not useful to use abbreviations for single words, like in this line (it doesn’t lower the word count, but it can introduce confusion for the reader). Other example is LEN (Lenvatinib)

Answer: We agree and thank the reviewer for the suggestion. We have replaced all acronyms LEN and SOF with their extended form.

Round 2

Reviewer 1 Report

Comments were addressed. Thank you.